# Don't Judge by the Look: Towards Motion Coherent Video Representation

**Yitian Zhang**[1][*] **Yue Bai**[1] **Huan Wang**[1] **Yizhou Wang**[1] **Yun Fu**[1,2]
[1]Department of Electrical and Computer Engineering, Northeastern University
[2]Khoury College of Computer Science, Northeastern University

## Abstract

Current training pipelines in object recognition neglect Hue Jittering when doing data augmentation as it not only brings appearance changes that are detrimental to classification, but also the implementation is inefficient in practice. In this study, we investigate the effect of hue variance in the context of video understanding and find this variance to be beneficial since static appearances are less important in videos that contain motion information. Based on this observation, we propose a data augmentation method for video understanding, named Motion Coherent Augmentation (MCA), that introduces appearance variation in videos and implicitly encourages the model to prioritize motion patterns, rather than static appearances. Concretely, we propose an operation SwapMix to efficiently modify the appearance of video samples, and introduce Variation Alignment (VA) to resolve the distribution shift caused by SwapMix, enforcing the model to learn appearance invariant representations. Comprehensive empirical evaluation across various architectures and different datasets solidly validates the effectiveness and generalization ability of MCA, and the application of VA in other augmentation methods. Code is available at https://github.com/BeSpontaneous/MCA-pytorch.

## 1 Introduction

Video understanding has evolved rapidly due to the increasing number of online videos. Even though, current methods still suffer from the overfitting issue. For instance, TSM Lin et al. (2019) demonstrates a substantial disparity between its training (78.46%) and validation accuracy (45.63%) on Something-Something V1 Goyal et al. (2017) dataset with a notably high Expected Calibration Error (ECE) Guo et al. (2017) of 23.25%, indicating its tendency for overconfident predictions. A plausible explanation is that video understanding benchmarks usually have a smaller pool of training samples compared with object recognition datasets, e.g., ImageNet Deng et al. (2009) with 1.2 million training samples compared to Kinetics400 Kay et al. (2017) with 240K videos.

To alleviate overfitting, there are many data augmentation methods Zhang et al. (2017); DeVries & Taylor (2017); Yun et al. (2019); Cubuk et al. (2020) being designed to conduct various transformations to training samples, and current state-of-the-art methods Li et al. (2022a;b) in video understanding often incorporate multiple data augmentation operations for better generalization ability. Among them, Color Jittering is widely used to transform color attributes like saturation, brightness, and contrast. However, we notice that Hue Jittering, which changes the attribute of hue in color, is often overlooked in current object recognition Liu et al. (2022b; 2021) methods and the reasons can be concluded in two folds: first, Hue Jittering is thought to be detrimental to object recognition as the effect of hue variance looks like changing the appearance of objects which will increase the difficulty in classification; second, the current implementation of Hue Jittering Paszke et al. (2019) is highly time-consuming as it will transform the samples from RGB to HSV space back and forth.

The effect of Hue Jittering in video understanding has not been systematically studied in prior works. To examine its efficacy, we conduct experiments on object recognition dataset CIFAR10 Krizhevsky et al. (2009) with ResNet-56 He et al. (2016) and video recognition dataset Something-Something V1 Goyal et al. (2017) with TSM Lin et al. (2019). Fig. 1 shows that Hue Jittering results in

---

[*]Corresponding Author: markcheung9248@gmail.com.

Figure 1: Illustration of the effect caused by Hue Jittering (denoted by *). Hue Jittering will lead to unrealistic and confusing appearances which are detrimental to object recognition. However, it can improve the performance in video recognition where static appearances are less important.

an accuracy drop on CIFAR10, as hue variance will lead to unrealistic or confusing appearances of objects. For example, given an image of a '*Polar Bear*', it might be misclassified to '*Ursus arctos*' (brown bear) with hue variation. In contrast, the performance of video recognition method is improved, as its goal is to categorize the action in the given video, where static appearances are less important or even misleading. From Fig. 1, one can see that the action '*Catch*' in the generated video remains unaffected despite the hands and football turning into green color with Hue Jittering.

Despite its effectiveness in video understanding, the current implementation of Hue Jittering Paszke et al. (2019) still suffers from inefficiency because of the transformation between RGB and HSV space. To address the issue, we propose an efficient operation SwapMix to modify the appearance of video samples in the RGB space. Specifically, we generate videos with new appearances by randomly permuting the RGB order, while ensuring that other color attributes such as saturation and lightness will remain unchanged. However, swapping the channel order can only lead to discrete variations and the generated video may follow a fixed pattern. Thus, we further mix the original video and the generated video with linear interpolation to enlarge the input space. In this manner, we can efficiently alter the hue value of video samples and simulate the effect of appearance variation.

Nevertheless, it is worth noting that videos generated by SwapMix may exhibit unrealistic appearances, e.g., green-colored hands and football, which are unlikely to occur in the real world. Although SwapMix effectively enlarges the training set, the augmented samples will result in distribution shift compared to the original training set which may limit its performance. To address this issue, we propose Variation Alignment (VA) to construct training pairs with different appearances and encourage their predictions to align with each other. By providing the network with the prior knowledge of '*what is the relation between inputs with different variations*', we can enforce the model to learn appearance invariant representation, which is shown to be beneficial for video understanding.

Considering SwapMix and Variation Alignment (VA) in a uniform manner, we build Motion Coherent Augmentation (MCA), which efficiently generates video with new appearances and resolves the distribution shift caused by appearance variation. In this way, we can implicitly guide the model to prioritize the motion pattern in videos, instead of solely relying on static appearance for predictions. Notably, most training pipelines do not consider the effect of hue variance, which means that MCA is compatible with existing data augmentation approaches and can further improve the generalization ability of competing methods like Uniformer Li et al. (2022a). Moreover, we conduct experiments to validate that VA, as a plug-in module, can resolve the distribution shift of other data augmentation methods for even better performance. We summarize the contributions as follows:

- Despite its negative impact on object recognition, we reveal that Hue Jittering is beneficial in video understanding as appearance variance does not affect the action conveyed by video.
- We propose a data augmentation method Motion Coherent Augmentation (MCA) to learn the appearance invariant representation for video understanding. Concretely, we present SwapMix to efficiently simulate the effect of hue variance and introduce Variation Alignment (VA) to resolve the distribution shift caused by SwapMix.
- Comprehensive empirical evaluation across different architectures and benchmarks substantiates that MCA, which can be seamlessly integrated into established video understanding approaches with minimal code modifications, yields consistent improvement and demonstrates excellent compatibility with existing data augmentation techniques.
- Analysis of Variation Alignment (VA) validates that it can be utilized to resolve the distribution shift of other data augmentation methods for even better performance.

## 2 RELATED WORK

**Video Recognition** has benefited a lot from the development in object recognition, especially the 2D-based methods which share the same backbone with the models in object recognition. The research focus of these 2D methods lies in temporal modeling Wang et al. (2016); Lin et al. (2019); Li et al. (2020). Another line of research focuses on using 3D-CNNs to capture the spatial and temporal relation jointly, such as I3D Carreira & Zisserman (2017), C3D Tran et al. (2015) and Slow-Fast Feichtenhofer et al. (2019). Though effective, these methods usually cost great computations. Based on the structure of Vision Transformers Dosovitskiy et al. (2020), many Transformer-based networks Fan et al. (2021); Liu et al. (2022a); Li et al. (2022a) have been introduced for spatial-temporal learning in video recognition and exhibited impressive performance.

**Data Augmentation** has proven to be effective in mitigating the effect of overfitting. Traditional augmentation methods, including random flipping, resizing, and cropping, are frequently used to enforce invariance in deep networks He et al. (2016); Huang et al. (2017). Some studies have investigated to enlarge the input space by randomly occluding certain regions in images DeVries & Taylor (2017), blending two images Zhang et al. (2017), or replacing an image patch with the one in another image Yun et al. (2019). In recent years, automatic augmentation strategies Cubuk et al. (2018; 2020) are shown to be effective as they consider extensive augmentation types, such as color jittering, shear, translation, etc. Apart from those image-based approaches, recent works Li et al. (2023); Zou et al. (2023); Gowda et al. (2022); Kimata et al. (2022); Yun et al. (2020a); Wang et al. (2021) have made attempts to address the background bias issue in video recognition, allowing the model to concentrate more on the motion patterns in videos. While we do not focus on this particular problem, our method can partially address this issue as it will also cause hue variance in the background area and help the model to rely less on the foreground bias information as well.

**Knowledge Distillation** is proposed to train a student network to mimic the behavior of a larger teacher model Hinton et al. (2015). To avoid the extra costs of teacher network in previous methods Park et al. (2019); Ahn et al. (2019); Tian et al. (2019), researchers have developed self-distillation approaches that allow models to transfer their own knowledge into themselvesZhu et al. (2018); Xu & Liu (2019); Yun et al. (2020b); Zhang et al. (2019). Among them, CS-KD Yun et al. (2020b) and data distortion Xu & Liu (2019) are relevant to our work as both of them construct training pairs and encourage similar predictions. However, CS-KD uses different training samples within the class to construct the training pair, and data distortion applies the same augmentation to both training samples. In contrast, our method mainly focuses on the appearance variation in videos and utilizes the same sample with different appearances to learn the invariant representations.

## 3 MOTION COHERENT AUGMENTATION

We first introduce the preliminaries of Affinity and Diversity Cubuk et al. (2021), two metrics that measure the distribution shift and uniqueness of the augmentation operations. Then, we present our method Motion Coherent Augmentation (MCA) which is shown in Fig. 2. Specifically, we introduce the operation of SwapMix which can efficiently result in hue variance and alter the appearance of the given videos. Further, we propose Variation Alignment (VA) to resolve the distribution shift caused by SwapMix. By encouraging the model to generate consistent predictions for videos with varying appearances, we can enforce the model to learn appearance invariant representations and prioritize the extraction of motion-related information.

### 3.1 PRELIMINARIES

Prior work Cubuk et al. (2021) has proposed two quantitative measures to analyze existing data augmentation methods. The first one is Affinity which describes how the augmentation operation shifts data with respect to the decision boundary, while the other one is Diversity which quantifies the uniqueness of the augmented training set.

Formally, given a video $x \in X$ containing T frames $x = \{f_1, f_2, ..., f_T\}$ with its one-hot class label $y \in Y$, we denote $Y = [0, 1]^K$ and $K$ is the number of classes. Assume a data augmentation operation $a(\cdot)$, the augmented video sample $\tilde{x}$ can be obtained:

$$\tilde{x} = a(x). \tag{1}$$

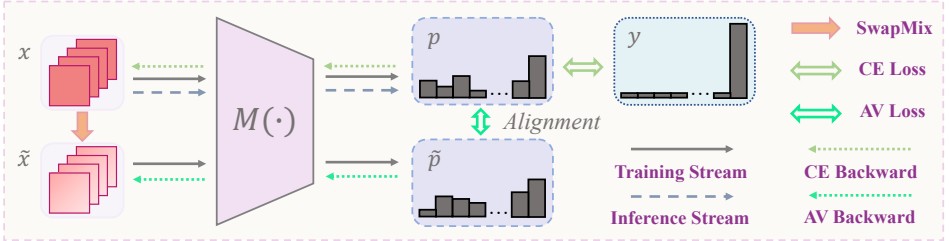

Figure 2: Illustration of Motion Coherent Augmentation (MCA). During training, given input video $x$, we will modify its appearance by SwapMix to get $\tilde{x}$. We feed the input pair into the deep network $M(\cdot)$ to obtain corresponding predictions $p$ and $\tilde{p}$. $p$ will be used to calculate CE loss and $\tilde{p}$ will be encouraged to align with $p$ to enforce the model to learn appearance invariant representation and resolve the distribution shift caused by SwapMix. During inference, only $x$ will be used.

Let $X_{train}$ and $X_{val}$ be the training and validation sets drawn IID from the clean data distribution $X$, we denote $\tilde{X}_{val}$ which is derived from $X_{val}$ by applying $a(\cdot)$ to each video in $X_{val}$:

$$\tilde{X}_{val} = \{(a(x), y) : \forall (x, y) \in X_{val}\}. \tag{2}$$

Suppose $M(\cdot)$ to be a model trained on $X_{train}$ and $\Lambda(M, X_{val})$ represents the accuracy of the model when evaluated on dataset $X_{val}$, the Affinity of augmentation operation $a(\cdot)$ can be defined as:

$$\tau[a; M; X_{val}] = \Lambda\left(M, \tilde{X}_{val}\right) / \Lambda(M, X_{val}). \tag{3}$$

$\tau[a; M; X_{val}] = 1$ implies that there is no distribution shift and a smaller number indicates the larger distribution shift caused by the augmentation operation. Similarly, we define $\tilde{X}_{train}$ by applying augmentation $a(\cdot)$ to each video in $X_{train}$ and we further denote the train loss on $X_{train}$ over the model $M(\cdot)$ as $L_{train}$ so that Diversity can be written as:

$$D[a; M; X_{train}] = \tilde{L}_{train} / L_{train}, \tag{4}$$

where higher Diversity suggests that there is a large variance in the training samples. Typically, an ideal data augmentation operation should exhibit high Diversity which brings more variance to the training samples, and high Affinity to avoid huge distribution shift.

## 3.2 SwapMix

Considering a pixel in an RGB format image, we represent the values of RGB channels as $r$, $g$, $b$, and max, min stand for the maximum and minimum values of the three. At first, Hue Jittering will transform the pixel to HSV space based on:

$$H = \begin{cases} 0°, & \max = \min, \\ 60° \times \frac{g-b}{\max - \min} + 0°, & \max = r \,\&\, g \geqslant b, \\ 60° \times \frac{g-b}{\max - \min} + 360°, & \max = r \,\&\, g < b, \\ 60° \times \frac{b-r}{\max - \min} + 120°, & \max = g, \\ 60° \times \frac{r-g}{\max - \min} + 240°, & \max = b, \end{cases} \quad S = \begin{cases} 0, & \max = 0, \\ \frac{\max - \min}{\max}, & \max \neq 0, \end{cases} \quad V = \max, \tag{5}$$

where $H$ represents hue, $S$ stands for saturation, and $V$ denotes value. Then, Hue Jittering will adjust hue in the HSV space and map the altered value to RGB space which is inefficient in practice.

We can observe from Eq. 5 that $S$, $V$ are only determined by the maximum and minimum values of the RGB channels, and the value of $H$ can be easily changed by shuffling the orders of the RGB channels without affecting $S$ and $V$. Considering an RGB frame $f_i \in \mathbb{R}^{\left(C_i^R, C_i^G, C_i^B\right) \times H \times W}$ with 3 channels in total, the video sequence can be represented as $x \in \mathbb{R}^{T \times \left(C^R, C^G, C^B\right) \times H \times W}$. We can simulate the effect of hue variance by the operation $a_{CS}(\cdot)$, which randomly permutes the RGB channels. The generated video $x'$ can be written as:

$$x' = a_{CS}(x), \quad f_i' \in \mathbb{R}^{\left(C_i^{\phi[1]}, C_i^{\phi[2]}, C_i^{\phi[3]}\right) \times H \times W}, \tag{6}$$

where $\phi \in \{(RBG), (BRG), (BGR), (GRB), (GBR)\}$ and the sampling procedure is random. In this manner, we can efficiently create video $x'$ with a different appearance and we denote the operation $a_{CS}(\cdot)$ as Channel Swap.

Nevertheless, Channel Swap can only lead to discrete variance and this behavior may not be optimal because the generation follows a fixed pattern. As shown in Fig. 3, we interpolate between $x$ and $x'$ to generate the augmented sample $\tilde{x}$ with a continuous range of appearance variation and the coefficient $\lambda$ is drawn from the Beta distribution $Beta\left(\alpha, \alpha\right)$ to control the strength. In our implementation, we set $\alpha$ to 1 so $\lambda$ will be sampled from the uniform distribution $(0, 1)$. We define the operation of SwapMix which generates training sample $\tilde{x}$ and the label as:

$$\left\{ \begin{array}{l} \tilde{x} = \lambda x + \left(1 - \lambda\right) x', \\ \tilde{y} = y. \end{array} \right. \qquad (7)$$

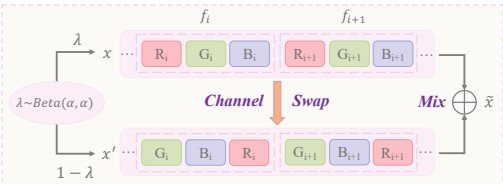

Figure 3: Illustration of SwapMix. Given a video $x = \{f_1, f_2, ..., f_T\}$, its channel order will be shuffled to create the video with a new appearance $x'$. Then, interpolation between $x$ and $x'$ will be implemented to generate $\tilde{x}$ with an enlarged input space. The coefficient $\lambda$ is sampled from the Beta distribution to control the degree of interpolation.

Due to the introduced $\lambda$, the degree of appearance variance can be measured with a linear relation, leading to continuous variance in the input space. We empirically compare the running time of SwapMix and Hue Jittering on different platforms in Fig. 4 and the results suggest that our method is way more efficient in practice.

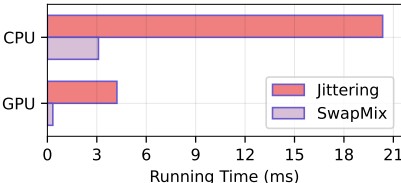

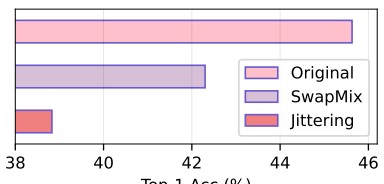

Figure 4: Running time comparisons of Hue Jittering and SwapMix on CPU (Intel(R) Core(TM) i7-6850K) and GPU (NVIDIA GeForce GTX TITAN X). Results are averaged over 500 runs.

Figure 5: Evaluation of TSM Lin et al. (2019) on different Something-Something V1 validation sets, where the data is original videos, and videos generated by SwapMix, Hue Jittering.

### 3.3 VARIATION ALIGNMENT

Though SwapMix can change the appearance of video samples which enlarges the input space and mitigates overfitting, the appearance of the generated video $\tilde{x}$ may differ significantly from the original data which results in distribution shift and low Affinity. Following the procedure in Sec. 3.1, we empirically measure the accuracy of TSM Lin et al. (2019) when evaluated on validation sets that are generated by different augmentation operations in Fig. 5. One can observe that SwapMix leads to less performance decline compared to Hue Jittering and the reasons can be summarized as: (1) Channel Swap does not change the distribution inside the channels as we only shuffle the channel orders, but Hue Jittering will alter the values of pixels because of the non-linear transformation across RGB and HSV spaces. (2) The interpolation results in a continuous input space which preserves more information about the original data.

Nonetheless, both augmentation operations suffer from tremendous performance drops which implies that the distribution shift caused by appearance variance is non-negligible. To address this issue, we propose Variation Alignment (VA) which constructs pairs of videos with different appearances, and explicitly enforces the network to learn the appearance invariant representations. Shown in Fig. 2, given video sample $x$, we can obtain its predictions $p$ by:

$$p = M\left(x; \theta\right), \qquad (8)$$

where $M\left(\cdot\right)$ is the deep network to extract features and $\theta$ is the parameters. Meanwhile, we will transform $x$ to $\tilde{x}$ by SwapMix and get its corresponding prediction $\tilde{p}$ by passing it through $M\left(\cdot\right)$ as well. By calculating Cross-Entropy loss on $p$, we obtain:

$$\mathcal{L}_{CE} = -\sum_{k=1}^{K} y_k \log\left(p_k\right), \qquad (9)$$

where $y_k$ represents the one-hot label of class $k$. Traditional data augmentation methods calculate the loss of the whole training set by Eq. 9 which inevitably introduces distribution shift and leads to

non-optimal performance. Instead of optimizing $\tilde{p}$ towards a specified one-hot label, we enforce the predictions of $p$ and $\tilde{p}$ to align with each other by minimizing the appearance variation loss:

$$\mathcal{L}_{AV} = -\sum_{k=1}^{K} p_k \log \left( \frac{\tilde{p}_k}{p_k} \right). \tag{10}$$

In this manner, we encourage the network to learn the appearance invariant representations by providing it with the prior knowledge of '*what is the relation between inputs with different variations*', rather than '*what is the right prediction for each input*'.

Combining the two losses together, we update the parameters $\theta$ in $M(\cdot)$ by:

$$\mathcal{L} = \mathcal{L}_{CE} + \lambda_{AV} \cdot \mathcal{L}_{AV}, \tag{11}$$

where $\lambda_{AV}$ is introduced to balance the two terms. In this way, we can resolve the distribution shift caused by SwapMix and implicitly encourage the model to focus more on the motion information, rather than solely relying on static appearance.

## 4 EMPIRICAL VALIDATION

We empirically evaluate the performance of Motion Coherent Augmentation (MCA) on various architectures and benchmarks in this segment. We first validate the effectiveness of MCA on different methods, datasets, and frames. Second, we present the comparisons of our method with other competing data augmentation methods and demonstrate that MCA is compatible with them. Further, we analyze the effect of Variation Alignment (VA), the training process, and robustness against probability change. We provide comprehensive ablation and qualitative analysis in the end.

### 4.1 EXPERIMENTAL SETUP

**Datasets.** We validate our method on five video benchmarks: (1) Something-Something V1 & V2 Goyal et al. (2017) are made up of 98k videos and 194k videos samples, which exhibit significant temporal dependencies and are employed for the majority of evaluations. (2) UCF101 Soomro et al. (2012) comprises 13,320 videos across 101 categories and the first training/testing split is adopted for training and evaluation. (3) HMDB51 Kuehne et al. (2011) consists of 6,766 videos categorized into 51 classes. Similarly, we employ the first training/testing split for both training and testing. (4) Kinetics400 Kay et al. (2017) is a large-scale dataset which is categorized into 400 action classes.

**Implementation details.** We sample 8 frames uniformly for all methods except for SlowFast Feichtenhofer et al. (2019) which samples 32 frames for fast pathway. During training, we crop the training data randomly to $224 \times 224$, and we abstain from applying random flipping to the Something-Something datasets. In the inference phase, frames will be center-cropped to $224 \times 224$ except SlowFast which is cropped to $256 \times 256$. We adopt **one-crop one-clip** per video during evaluation for efficiency unless specified. More implementation details can be found in the appendix.

### 4.2 MAIN RESULTS

**Evaluation across different datasets.** In this part, we validate Motion Coherent Augmentation (MCA) across different datasets in Tab. 1. As for UCF101 and HMDB51, the improvements brought by MCA are, quite obviously, greater than 2% as the scales of these datasets are relatively small, leading to a more pronounced occurrence of overfitting. Something-Something V2 is a large-scale video recognition dataset with strong temporal dependency and MCA can effectively increase the generalization ability of the baseline method by implicitly encouraging the model to prioritize the motion patterns during training. We further implement MCA on Kinetics400 which is a large-scale dataset and MCA still improves the performance of the baseline model. Note that MCA is still shown to be effective on datasets like UCF101 which contain less motion information, indicating the effectiveness of appearance invariant representations in video understanding.

**Evaluation across different architectures.** In Tab. 2, we empirically validate the performance of MCA across different architectures on Something-Something V1 dataset which exhibits obvious temporal dependency. We first implement MCA on 2D-nework TSM Lin et al. (2019) and one can

Table 1: Evaluation of Motion Coherent Augmentation (MCA) on UCF101, HMDB51, Something-Something V2 and Kinetics400. The best results are bold-faced.

| Method | UCF101 | | HMDB51 | | Something-Something V2 | | Kinetics400 | |
|---|---|---|---|---|---|---|---|---|
| | Acc1.(%) | Δ Acc1.(%) | Acc1.(%) | Δ Acc1.(%) | Acc1.(%) | Δ Acc1.(%) | Acc1.(%) | Δ Acc1.(%) |
| TSM | 79.57 | +2.30 | 48.63 | +3.00 | 59.29 | +1.42 | 70.28 | +0.80 |
| TSM+MCA | **81.87** | | **51.63** | | **60.71** | | **71.08** | |

Table 2: Evaluation of MCA on Something-Something V1 dataset with different architectures, including 2D, 3D and Transformer-network. The best results are bold-faced.

| Method | Acc1.(%) | Acc5.(%) | Δ Acc1.(%) |
|---|---|---|---|
| TSM | 45.63 | 75.00 | +1.94 |
| TSM+MCA | **47.57** | 76.27 | |
| SlowFast | 44.12 | 72.58 | +1.76 |
| SlowFast+MCA | **45.88** | 74.06 | |
| Uniformer | 48.48 | 76.69 | +2.03 |
| Uniformer+MCA | **50.51** | 78.20 | |

Table 3: Comparisons with competing data augmentation methods on Something-Something V1 dataset. The best results are bold-faced.

| Method | Acc1.(%) | Δ Acc1.(%) |
|---|---|---|
| TSM | 45.63 | - |
| TSM+Cutout | 44.68 | −0.95 |
| TSM+CutMix | 44.99 | −0.64 |
| TSM+VideoMix | 45.62 | −0.01 |
| TSM+Mixup | 46.03 | +0.40 |
| TSM+AugMix | 46.03 | +0.40 |
| TSM+BE | 46.45 | +0.82 |
| TSM+RandAugment | 47.21 | +1.58 |
| TSM+MCA | **47.57** | +1.94 |

observe that MCA significantly improves its performance by 1.94%. Further, we extend MCA to 3D-network: SlowFast Feichtenhofer et al. (2019) and Transformer-network: Uniformer-S Li et al. (2022a). The results show that MCA consistently increases the accuracy of these methods which validates that the idea of learning appearance invariant representation is effective and has great generalization ability across different architectures.

**Evaluation across different frames on competing method.** We further validate the performance of MCA across various frames on the state-of-the-art method Uniformer Li et al. (2022a). From Fig. 6, one can notice that MCA leads to consistent improvement at all frames with the mean value of 1.95% and the standard deviation is only 0.25 which demonstrates the stability of MCA. Moreover, as current state-of-the-art works normally adopt multi-crop evaluation to pursue better performance, we evaluate our method with the same procedure in Uniformer Li et al. (2022a) and present them in dotted lines. Similarly, one can observe similar enhancements across all these frames which implies that MCA can also result in improvement even on state-of-the-art works.

## 4.3 COMPARISON WITH OTHER APPROACHES

**Quantitative comparison.** We compare our method with competing data augmentation methods on Something-Something V1 based on TSM in Tab. 3. It is shown that popular data augmentation methods like Cutout DeVries & Taylor (2017), CutMix Yun et al. (2019) result in negative effects in video understanding. One possible explanation is that these methods break the motion pattern in video sequences and bring challenges to temporal modeling which is extremely essential in video understanding. Mixup Zhang et al. (2017) shows improvement as it enforces a regularization effect which is helpful to deal with overfitting. VideoMix Yun et al. (2020a) leads to similar performance with baseline method and Background Erasing (BE) Wang et al. (2021) is helpful due to the stress of motion information. AugMix Hendrycks et al. (2019) and RandAugment Cubuk et al. (2020) exhibit great performance as they involve multiple augmentation operations and their magnitudes will be adaptively sampled. MCA outperforms all these approaches even though we only consider the effect of hue variance in videos, which proves the efficacy of our method.

**Compatibility with other approaches.** Current state-of-the-art video understanding methods are usually incorporated with multiple strong data augmentation methods. For example, the official implementation of Uniformer-S Li et al. (2022a) intrinsically involves CutMix Yun et al. (2019), Mixup Zhang et al. (2017) and RandAugment Cubuk et al. (2020) and the improvement brought by MCA in Tab. 2 is made upon these methods. This indicates that our method is compatible with existing works as MCA only considers hue variance in videos which is overlooked in previous approaches. To better examine the effect of MCA, we remove other augmentations in Uniformer and build our method on top of it. From Fig. 7, we can see that MCA can lead to even more improvement in accuracy when we remove other augmentations, which demonstrates the strength of our method.

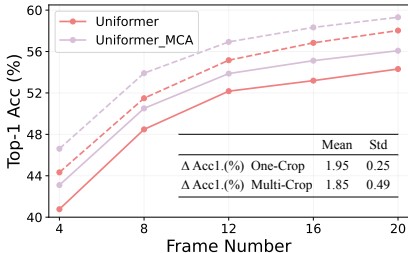

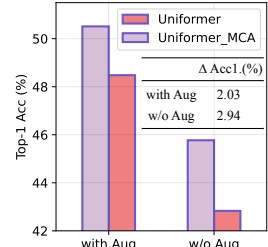

Figure 6: Evaluation of MCA at different frames over competing method Uniformer-S on Something-Something V1. Solid, dotted lines denote one-crop and multi-crop evaluation, respectively.

Figure 7: Compatibility of MCA with other competing data augmentation methods (include RandAugment, Mixup, Cut-Mix) on Something-Something V1.

Table 4: Extending Variation Alignment (VA) to competing data augmentation methods on Something-Something V1. The best results are bold-faced.

| Method | VA | Train Acc1.(%) | Val Acc1.(%) |
|---|---|---|---|
| TSM | - | 78.46 | 45.63 |
| TSM+AugMix | ✗ | 73.20 | 46.03 |
| TSM+RandAugment | ✗ | 63.94 | 47.21 |
| TSM+AugMix | ✔ | 81.53 | **46.55**(0.52↑) |
| TSM+RandAugment | ✔ | 81.55 | **48.53**(1.32↑) |

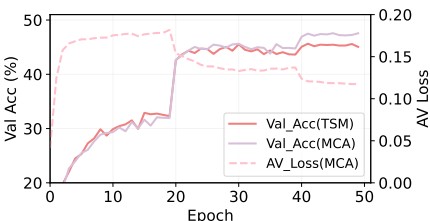

Figure 8: The curves of Validation Accuracy and AV Loss during training on Something-Something V1 dataset.

## 4.4 ABLATION AND ANALYSIS

**Analysis of Variation Alignment.** The motivation of VA is to alleviate the distribution shift resulting from SwapMix and encourage the model to learn representations invariant to appearance. Similarly, this idea can easily be extended to address the distribution shift of other augmentation methods. Shown in Tab. 4, when we extend VA to AugMix Hendrycks et al. (2019) and RandAugment Cubuk et al. (2020), we can further improve their performance by 0.52% and 1.32%. The results prove the generalization ability of our design and imply that VA can easily be applied to current augmentation methods to mitigate the distribution shift and result in even better performance. Besides, we notice a significant increase in the training accuracy when the methods are incorporated with VA, which suggests that VA enables the model to learn better representations.

**Analysis of Training Process.** We plot the curves of validation accuracy and AV loss in Fig. 8 based on the experiments of TSM Lin et al. (2019). One can notice that the accuracy will increase sharply at Epoch 20 and 40 because of the learning rate decay schedule. After Epoch 20, our method outperforms TSM as the AV loss starts to decrease after that, indicating that the model starts to learn the appearance invariant representations which is beneficial for the performance.

**Robustness against probability change.** Compared to the baseline method, our method will construct a training pair and enforce appearance invariant learning at each iteration. We introduce a hyperparameter $\rho$ and the model will apply the operation of MCA at this iteration if the value, randomly drawn from a uniform distribution, is less than the predetermined probability $\rho$. To study the robustness of MCA against probability change, we empirically conduct experiments over different $\rho$ and the results are in Tab. 5. The first observation is that MCA clearly outperforms the baseline method with a smaller ECE Guo et al. (2017), i.e., the difference between predicted probabilities and their true accuracy, under different $\rho$, which implies that models trained with MCA are more well-calibrated. Besides, there is only a small accuracy drop when we decrease the probability to 0.25 which means that our method is still effective even when we change the appearance of a limited number of data. Typically, applying data augmentation operations during training will result in longer training hours to different extents, and so does our method. This phenomenon can help us to alleviate this problem, as we can choose a small probability of applying MCA if we want to attain a better trade-off between training efficiency and validation accuracy.

**Ablation of design choices.** In this part, we provide ablations of design choices to verify our method in Tab. 6. We first implement Hue Jittering which changes the hue of videos with the official

Table 5: Robustoness against probability change on Something-Something V1. The best results are bold-faced.

| Method | $\rho$ | ECE.(%) | Acc1.(%) |
|---|---|---|---|
| TSM | - | 23.25 | 45.63 |
| TSM+MCA | 0.25 | 22.04 | 47.26 |
| TSM+MCA | 0.50 | 21.85 | 47.38 |
| TSM+MCA | 0.75 | **21.09** | **47.60** |
| TSM+MCA | 1.00 | 21.11 | 47.57 |

Table 6: Ablation of design choices on Something-Something V1. The best results are bold-faced.

| Method | Specification | Acc1.(%) |
|---|---|---|
| TSM+HJ | Hue Jittering | 46.32 |
| TSM+CS | Channel Swap | 45.98 |
| TSM+SwapMix | Remove VA from MCA | 46.53 |
| TSM+SwapMix | Expand Training Set | 45.69 |
| TSM+MCA | Add CE Loss on $\tilde{p}$ | 46.89 |
| TSM+MCA | - | **47.57** |

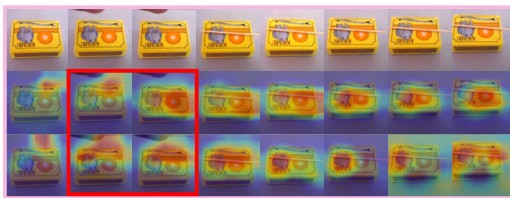

(a) Dropping something onto something.

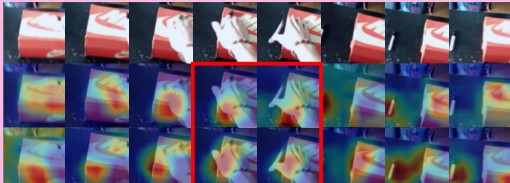

(b) Pushing something off of something.

Figure 9: Visualizations produced by CAM Zhou et al. (2016) on Something-Something V1 with TSM Lin et al. (2019) (second row) and TSM+MCA (third row). TSM misclassifies the first sample as '*Putting something onto something*' and the second sample as '*Hitting something with something*'.

implementation provided by Pytorch Paszke et al. (2019). It exhibits worse performance compared to SwapMix with a more inefficient operation. Further, we implement Channel Swap and SwapMix, and results show that both methods can lead to improvement over the baseline. However, their performance is restrained by the distribution shift caused by appearance variation. The performance of SwapMix is better than Channel Swap as the interpolation offers a continuous distribution of the training data which enlarges the input space. Further, we explicitly expand the training set by creating training pairs like MCA and calculating CE loss on all the video samples to update the parameters. The result is even worse than SwapMix which indicates that the improvement brought by MCA does not come from the larger training set. Finally, we add CE loss on prediction $\tilde{p}$ and the inferior performance suggests that computing CE loss on augmented training samples will introduce the distribution shift again and it indeed limits the performance of SwapMix.

**Qualitative analysis.** To demonstrate that MCA can encourage the model to concentrate more on the motion patterns, we utilize CAM Zhou et al. (2016) to visualize the attention maps produced by TSM Lin et al. (2019) and TSM+MCA on Something-Something V1 in Fig. 9. TSM misclassifies the first sample as '*Putting something onto something*' because it entirely concentrates on the yellow box at all the frames. While MCA helps the model to focus on the movement in the first three frames and make the right prediction. In the second case, the baseline model detects the action of the hand, but it places more emphasis on the wrist and overlooks the pen movement. In contrast, MCA directs the model to concentrate more precisely on the action of push.

## 5 CONCLUSION AND LIMITATIONS

In this work, we reveal that hue variance, which is thought to be detrimental to object recognition as it changes the appearance of images, is beneficial in video understanding. Based on the observation, we propose Motion Coherent Augmentation (MCA) to encourage the model to prioritize motion information, rather than static appearances. Specifically, we propose SwapMix to efficiently transform the appearance of video samples and leverage Variation Alignment (VA) to resolve the distribution shift caused by SwapMix, enforcing the model to learn appearance invariant representations. Comprehensive evaluation validates the effectiveness and generalization ability of MCA, and the application of VA in other data augmentation methods for even better performance.

One limitation of MCA is its increased demand for GPU memory throughout the training phase as we will import another batch of data with different appearances. Second, MCA will cause longer training time because we need to construct training pairs to calculate AV loss. In subsequent research, we aim to enhance the training efficiency.

ETHICS STATEMENT

In our paper, we strictly follow the ICLR ethical research standards and laws. To the best of our knowledge, our work abides by the General Ethical Principles.

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

# A   APPENDIX

## A.1   IMPLEMENTATION DETAILS

All models are trained on NVIDIA Tesla V100 GPUs with the same training hyperparameters as the official implementations. We set the hyperparameter $\lambda_{AV} = 1$ to be $1, 0.65, 0.4$ on Something-Something V1, V2 Goyal et al. (2017), Kinetics400 Kay et al. (2017) datasets, respectively. The default probability of applying Motion Coherent Augmentation (MCA) at each iteration is $p = 1$ in order to strive for better performance.

## A.2   VALIDATION IN SELF-SUPERVISED LEARNING

Table 7: Evaluation of MCA in self-supervised learning on UCF101 dataset. The best results are bold-faced.

| Method | UCF101 | |
|---|---|---|
| | Acc1.(%) | $\Delta$ Acc1.(%) |
| MoCo He et al. (2020) | 63.68 | +4.31 |
| MoCo+MCA | **67.99** | |

To demonstrate the effectiveness of MCA in self-supervised learning, we implement MoCo He et al. (2020) on UCF101 Soomro et al. (2012) and further combine it with MCA. Specifically, we conduct pre-training and fine-tuning both on UCF101 following the setting of Background Erasing (BE) Wang et al. (2021) and utilize the 3D network I3D Carreira & Zisserman (2017) as the base encoder. Further, we reduce the resolution during pre-training to $112 \times 112$ and increase the batch size for efficiency. One can observe from Tab. 7 that MCA leads to significant improvement over MoCo on the UCF101 dataset which proves the efficacy and generalization ability of MCA in different video understanding tasks.

## A.3   TRANSFERABILITY

Table 8: Evaluation of MCA with linear probing on HMDB51 dataset. The models are pre-trained on Kinetics400 with different data augmentation approaches. The best results are bold-faced.

| Method | Acc1.(%) |
|---|---|
| TSM Lin et al. (2019) | 63.59 |
| TSM+AugMix | 64.18 |
| TSM+RandAugment | 65.10 |
| TSM+MCA | **65.95** |

In order to prove the representation learned by MCA can be generalized to other domains, we first pre-train TSM Lin et al. (2019) on Kinetics400 Kay et al. (2017) with different data augmentation methods: AugMix Hendrycks et al. (2019) and RandAugment Cubuk et al. (2020), and then conduct linear probing on HMDB51 Kuehne et al. (2011) dataset to compare the downstream performance. As shown in Tab. 8, one can notice that MCA results in the highest accuracy compared to other methods, which proves that the representation learned by MCA is general and can be transferred to other domains.

## A.4   HYPERPARAMETER STUDY

We further conduct ablation with different $\lambda_{AV}$ over TSM Lin et al. (2019) on Something-Something V1 Goyal et al. (2017) dataset in this section. One can observe from Tab. 9 that MCA leads to consistent improvements with different choices of $\lambda_{AV}$ and MCA is robust to this hyperparameter variation. However, if $\lambda_{AV}$ is too large or small, we observe that the performance will start to decrease which meets our expectation as MCA will degenerate to the baseline if $\lambda_{AV} = 0$ and $\mathcal{L}_{AV}$ will dominate the training if $\lambda_{AV}$ is too large.

Table 9: Evaluation of MCA with different $\lambda$. The best results are bold-faced.

| Method | $\lambda$ | Acc1.(%) |
|---|---|---|
| TSM Lin et al. (2019) | - | 45.63 |
| TSM+MCA | 0.25 | 46.79 |
| TSM+MCA | 0.50 | 47.40 |
| TSM+MCA | 1.00 | **47.57** |
| TSM+MCA | 2.00 | 47.03 |
| TSM+MCA | 3.00 | 46.57 |

## A.5 HUE VARIANCE VERSUS BACKGROUND SCENE VARIANCE

In this part, we discuss the relation between our method MCA, and methods that try to address the background bias issue. First, 'video appearance' is a general concept that contains many attributes, such as color and background scene information. The motivation of MCA is to cause hue variance in training samples which leads to appearance changes in videos so that MCA can force the model to learn appearance invariant representations. For methods that target the background issue, they aim to cause background scene variation in videos which will also result in appearance changes and encourage the model to learn representations invariant to appearance. Although starting from different perspectives, the goal of these two research directions is similar: learn appearance invariant representations and prioritize the motion information. Moreover, we highlight that these two lines of research can be combined with each other. Shown in Tab. 10, we combine our method with prior work Background Erasing (BE) Wang et al. (2021) that tries to address the background bias issue, and leads to further improvement when combined with it.

Table 10: Compatibility of MCA with BE. The best results are bold-faced.

| Method | Acc1.(%) |
|---|---|
| TSM Lin et al. (2019) | 45.63 |
| TSM+BE | 46.45 |
| TSM+MCA | 47.57 |
| TSM+BE+MCA | **48.05** |

## A.6 RESULTS OF DIFFERENT DEPTHS

In previous experiments, most of the results on TSM Lin et al. (2019) are built upon ResNet-50 He et al. (2016) and we further conduct experiments on ResNet-18 and ResNet-101 to verify the effectiveness of MCA when evaluated on models with different representation abilities. One can observe from Fig. 10 that MCA leads to consistent improvement at different depths, which demonstrates that our method is effective regardless of the representation ability of the model. Further, we notice that TSM+MCA on ResNet-50 outperforms TSM on ResNet-101 with much fewer parameters and computational costs, suggesting the strength of MCA.

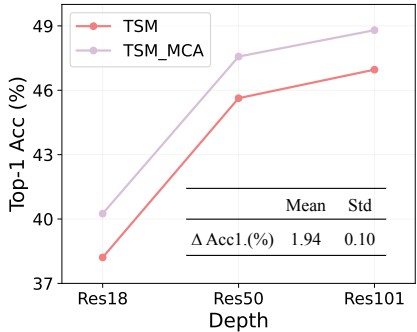

Figure 10: Experiments with varied levels of depth on Something-Something V1 dataset. The mean and standard deviation of Top-1 Accuracy improvement are shown in the table.

### A.7 Results on ActivityNet dataset

Table 11: Evaluation of MCA on ActivityNet dataset. The best results are bold-faced.

| Method | ActivityNet | |
| --- | --- | --- |
| | mAP(%) | $\Delta$ mAP(%) |
| TSM Lin et al. (2019) | 73.10 | +1.65 |
| TSM+MCA | **74.75** | |

The ActivityNet-v1.3 dataset Caba Heilbron et al. (2015) is an extensive collection of untrimmed videos, featuring 200 action categories and an average length of 117 seconds per video. It includes 10,024 video samples designated for training purposes and another 4,926 videos set aside for validation. From Tab 11, it can be observed that MCA still leads to a performance increase on the long video dataset which validates the effectiveness of our method.

### A.8 Analysis of Training Process

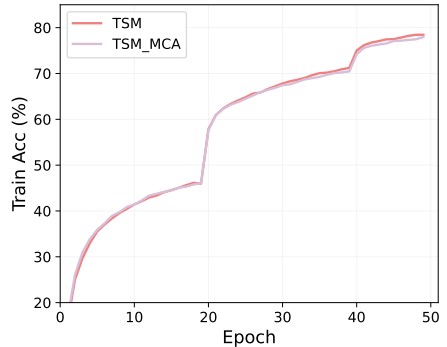 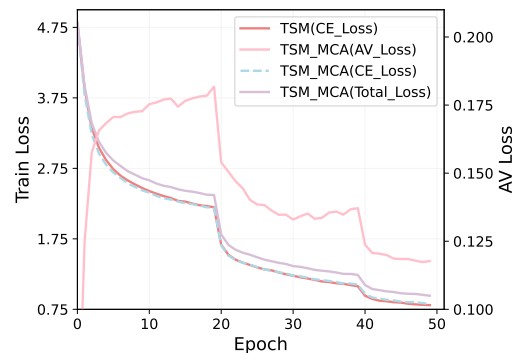

Figure 11: The curves of Training Accuracy during training on Something-Something V1.

Figure 12: The curves of Training Losses during training on Something-Something V1.

We have analyzed the training process in Sec. 4.3 and we further plot more training curves for analysis. We compare the training accuracy in Fig. 11, where it is noticeable that the two curves closely align with each other. While traditional data augmentation methods usually result in decreases in training accuracy to alleviate overfitting, our method introduces Variation Alignment (VA) which resolves the distribution shift caused by augmentation operations and can obtain similar training accuracy compared to the baseline method TSM Lin et al. (2019).

We further plot the curves of training losses of two models. The total loss of TSM+MCA is larger than the training loss of TSM because of the AV Loss introduced by VA, and we find that the curves of CE Loss of TSM+MCA and TSM are well-aligned with each other. With the decrease of AV Loss, the model is enforced to learn appearance invariant representation which is beneficial for video recognition.

### A.9 Qualitative Analysis

Here we show more visualization results on Something-Something V1 validation set in Fig. 13. We find that MCA can effectively lead the model to concentrate more on the motion patterns in videos and result in more accurate predictions.

### A.10 Training Time Analysis

As we mentioned, training efficiency is the limitation of MCA and we will analyze the training time in this section. Specifically, we measure the training time of MCA across different probabilities on 4 NVIDIA Tesla V100 GPUs and compare them with Hue Jittering (HJ) and Background Erasing (BE) Wang et al. (2021) which shares a similar vision with us to stress the motion patterns in videos.

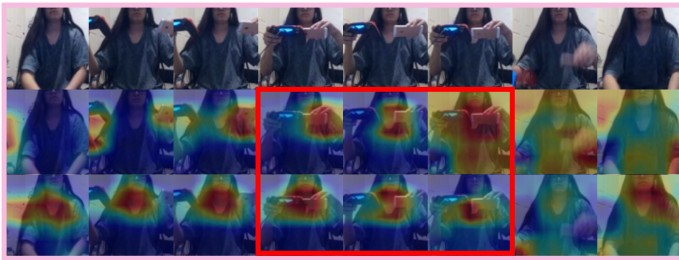

(a) Moving something and something so they collide with each other.

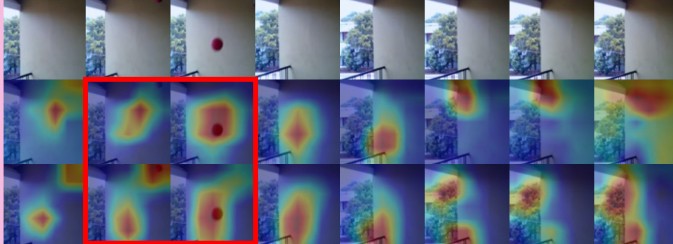

(b) Throwing something against something.

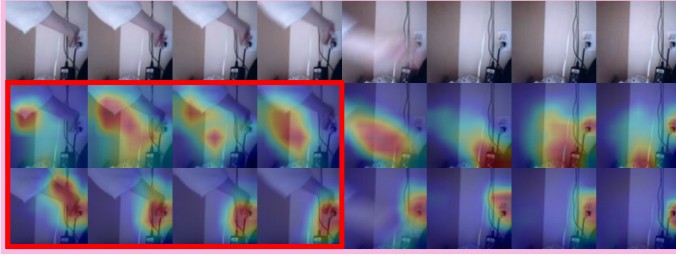

(c) Plugging something into something.

Figure 13: Visualizations produced by CAM Zhou et al. (2016) on Something-Something V1 with TSM Lin et al. (2019) (second row) and TSM+MCA (third row). TSM misclassifies the first sample as '*Attaching something to something*' and the second, the third sample as '*Something falling like a feather or paper*'.

One can see from Tab. 12 that MCA will inevitably increase the training time of the baseline method as other data augmentation methods. However, our method requires fewer training hours compared to HJ and BE when $\rho$ is smaller than 0.75 with a clear advantage in accuracy.

Meanwhile, it can be observed that the training time of MCA varies a lot with different values of $\rho$, while there is only a small change in accuracy. This finding can somehow mitigate this issue, as we can choose a small probability of applying MCA if we want to attain a better trade-off between training efficiency and validation accuracy.

Table 12: Training time of MCA across different probabilities on 4 NVIDIA Tesla V100 GPUs.

| Method | $\rho$ | Training Hours | Acc1.(%) |
|---|---|---|---|
| TSM | - | 10 | 45.63 |
| TSM+HJ | - | 15 | 46.32 |
| TSM+BE | - | 18 | 46.45 |
| TSM+MCA | 0.25 | 12 | 47.26 |
| TSM+MCA | 0.50 | 13 | 47.38 |
| TSM+MCA | 0.75 | 16 | 47.60 |
| TSM+MCA | 1.00 | 21 | 47.57 |

