# OpenReview forum: "Don't Judge by the Look: Towards Motion Coherent Video Representation"
_ICLR.cc/2024/Conference — ICLR 2024 poster_

### Official Review · Reviewer_7MY2 · 2023-10-20

**Soundness:** 2 fair
**Presentation:** 3 good
**Contribution:** 1 poor
**Rating:** 3
**Confidence:** 5

**Summary:**

This paper proposes a method of augmentation for video action recognition. The authors focus on “Hue Jittering”, which is one of common ways to data augmentation for images since it is implemented in common frameworks such as pytorch.

However it is computationally inefficient to compute hue and perform  jittering, then the authors propose SwapMix, which is randomly swap RGB channels of a given video frames (Channel Swap), followed by a linear blending with the original frames.

In addition, the authors propose Variation Alignment (VA) to enhance the invariant to appearance change by SwapMix. It is simply align the  model outputs for the original and SwapMix-ed frames by minimizing cross-entropy of two outputs.

Motion Coherent Augmentation (MCA), the name of the proposed method, is the combination is SwapMix and VA, which is used in experiments.

Experimental results demonstrated that
- the proposed MCA improved performances of different models (TSM, SlowFast, and Uniformer) for different datasets (SSv1/v2, UCF101, HMDB51),
- and outperformed other augmentation methods such as Cutout and CutMix.

**Strengths:**

It is interesting that changing hue of video frames is a pretty simple and channels swap is very fast to compute even on CPU, nevertheless the proposed MCA provides performance improvements on common action recognition datasets.
It might become a practical standard for endusers as it is easy to use and implement.

**Weaknesses:**

The main weakness of the paper is the lack of solid motivation.

Insufficient survey

The paper says “there has been a lack of studies for data augmentation methods in video recognition” in section 2, however, there a lot of works such as follows.

* Mitigating and Evaluating Static Bias of Action Representations in the Background and the Foreground, ICCV2023
* Learning representational invariances for data-efficient action recognition. 2023 CVIU
* Frequency Selective Augmentation for Video Representation Learning, AAAI 2023
* Enabling Detailed Action Recognition Evaluation Through Video Dataset Augmentation, NeurIPS 2022
* Learn2Augment: Learning to Composite Videos for Data Augmentation in Action Recognition, ECCV 2022
* ObjectMix: Data Augmentation by Copy-Pasting Objects in Videos for Action Recognition, MMAsia22
* Exploring Temporally Dynamic Data Augmentation for Video Recognition, arXiv 2022
* VideoMix: Rethinking Data Augmentation for Video Classification, arXiv 2020

Table 3 shows comparisons with other augmentation methods, however, which are for images. It should be better to compare other video augmentation methods for comparisons as the proposed method is for videos.

In addition, these prior works have tried to tackle the background bias or scene bias (similar to “static appearance” in page 2), which exists in action recognition datasets. It should be better to discuss how the proposed method can contribute to solve the issue.



Weak motivation for hue jittering: The paper says “Among them, Color Jittering is widely used to transform color attributes like saturation, brightness, and contrast” in the introduction. Indeed it is true, but there are a tens of methods for augmentations implemented in famous libraries such as imageaug, albumentations, and kornia. However, there is no discussions or evidences why hue jittering is so important to explore.

Weak motivation for backbone model: Experiments used TSM, a 3D CNN with shift modules, however it is merely one of a vast amount of models for action recognition. Other models (slowfast and uniformer) are used but not for all datasets. It should be justified why TSM is the best for demonstrating the effect of the proposed method. Otherwise, recent methods (such as ones  referred in the paper) should also compared.


Minor comments:

Channel swap: Swapping channels for augmentation is not new and have been implemented in libraries.
https://kornia.readthedocs.io/en/latest/augmentation.module.html#kornia.augmentation.RandomChannelShuffle
https://albumentations.ai/docs/api_reference/full_reference/#albumentations.augmentations.transforms.ChannelShuffle


Definitions: Affinity, Diversity, and high Expected Calibration Error (ECE) are mentioned in the paper, and only Affinity is defined. However, ECE only is shown in experiments (affinity may be from Fig5 indirectly).

**Questions:**

Questions are same as above; comparisons and motivations.

Figure 5 shows TSM with SwapMix, and the performance drops to about 42%. However in Table 5, TSM+SwapMix shows performance over 46%. Why this happens ?

---

> ### Author Response · Authors · 2023-11-15
> **Response to Reviewer 7MY2 (part 1)**
>
> We appreciate the Reviewer's feedback. We make further explanations to clarify the Reviewer's concerns based on several key points as below.
>
>
> **Weakness 1: Lack of motivation.**
>
> Thanks for the comment. To better clarify our motivation, we divide our motivation into two folds:
>
> (1) Why do we study the problem of Hue Jittering?
>
> As shown in Fig. 1, Hue Jittering is often neglected in current supervised video recognition methods because of its detrimental effect in object recognition. We rethink this problem in the context of video recognition and find the behavior of hue variance, which leads to appearance changes, is beneficial as static appearances are less important in videos. This justifies why MCA can lead to improvement on different datasets, architectures, and other data augmentation methods, as this problem generally exists in video recognition and is often neglected in current methods.
>
> (2) Why do we propose MCA instead of directly using Hue Jittering?
>
> The current implementation of Hue Jittering is highly time-consuming and its efficacy is limited by the distribution shift problem. To solve these two issues, we propose SwapMix to simulate the effect of hue variance while being significantly faster in practice, and then we present Variation Alignment to resolve the distribution shift and implicitly encourage the model to focus more on the motion information.
>
> The first point explains why our method is very general and the second point justifies the effectiveness of MCA.
>
> **Weakness 2: Insufficient survey and comparisons.**
>
> We thank the Reviewer for pointing out these works and we will add discussion with all of them in the final version. As for empirical comparisons, we stress that it is hard to fairly compare our method with most of them for the following reasons:
>
> <1> *'Mitigating and Evaluating Static Bias of Action Representations in the Background and the Foreground'*: This paper mainly targets the problem of out-of-distribution (OOD) generalization and it results in inferior performance on independent and identically distributed (IID) test data (shown in their Tab. 2/3/4), which means it leads to performance drop in the supervised learning setting.
>
> <2> *'Learning representational invariances for data-efficient action recognition'*: This work proposes ActorCutMix to swap the video background and needs a human detection model to generate human/background mask for each frame which relies on extra knowledge and is very time-consuming for implementation. Moreover, this paper mainly focuses on the semi-supervised learning setting which is different from us.
>
> <3> *'Frequency Selective Augmentation for Video Representation Learning'*: To our best knowledge, there is no publicly available code for the proposed method and this work mainly focuses on the self-supervised learning setting. We have tried to re-implement their method based on the pseudo-code, but errors appeared as the output is complex numbers.
>
> <4> *'Enabling Detailed Action Recognition Evaluation Through Video Dataset Augmentation'*: Instead of as a data augmentation method, the aim of this paper is to deliver an analysis toolkit for evaluation.
>
> <5> *'Learn2Augment: Learning to Composite Videos for Data Augmentation in Action Recognition'*: To our best knowledge, there is no publicly available code for the proposed method and it needs to train a selector using reinforcement learning.
>
> <6> *'ObjectMix: Data Augmentation by Copy-Pasting Objects in Videos for Action Recognition'*: To our best knowledge, there is no publicly available code for this method.
>
> <7> *'Exploring Temporally Dynamic Data Augmentation for Video Recognition'*: To our best knowledge, there is no publicly available code for this method.
>
> <8> *'VideoMix: Rethinking Data Augmentation for Video Classification'*: We implement the VideoMix on Something-Something V1 dataset over TSM[1] to compare with MCA and test different choices of hyperparameter as they do.
>
> | Method | $\alpha$ | Top-1 Acc. |
> | :-----: | :-----: | :-----: |
> | TSM | - | 45.63% |
> | TSM+VideoMix | 0.4 | 45.62% |
> | TSM+VideoMix | 2 | 45.25% |
> | TSM+VideoMix | 8 | 44.48% |
> | TSM+MCA | - | 47.57% |
>
> One can observe that VideoMix exhibits worse performance compared to the baseline under different hyperparameters, while MCA clearly improves the generalization ability of the baseline method which proves the efficacy of our method. Thanks again for listing these works and we will add discussion in our final version.

---

> ### Author Response · Authors · 2023-11-15
> **Response to Reviewer 7MY2 (part 2)**
>
> **Weakness 3: How MCA contributes to the background bias issue.**
>
> Thanks for the great question. Unlike prior works which divide each frame into foreground and background, we simply treat every image as a whole and change its appearance by SwapMix. That means we will also change the appearance of the foreground, which is usually human, as the variance in their looks does not affect the action conveyed in videos (shown in Fig. 1). By doing so, we can implicitly enforce the model to focus more on the motion information, rather than static appearances. This indicates that MCA is compatible with those works as our motivation is quite different from theirs. To validate it, we implement the prior work BE[2], which is robust to background bias, as a data augmentation operation and adapt it in supervised learning where we train it on Something-Something V1 dataset over TSM for comparisons:
> | Method | Top-1 Acc. |
> | :-----: | :-----: |
> | TSM | 45.63% |
> | TSM(BE) | 46.45%(+0.82) |
> | TSM(MCA) | 47.57%(+1.94) |
> | TSM(BE+MCA) | 48.05%(+2.42) |
>
> As shown in this table, MCA clearly outperforms BE and is compatible with BE which leads to significant improvement of the baseline method. It implies that MCA is compatible with those works and can contribute to further addressing the background bias issue.
>
> **Weakness 4: Weak motivation for Hue Jittering.**
>
> As we have explained in Weakness 1, Hue Jittering is often neglected in current supervised video recognition methods because of the detrimental effect in image classification and we find this operation is beneficial in the context of video recognition as static appearances are less important in videos. By exploring this direction, we deliver a simple but effective method MCA which has strong generalization ability and can be seamlessly integrated into existing approaches to improve their performance (shown in Tab. 1/2, Fig. 6/7).
>
> Although these libraries support this operation, they are inefficient in practice and that's why we propose SwapMix which can simulate this effect while being significantly faster. More importantly, the motivation is quite different as they simply treat it as a color attribute to increase the data diversity and we mainly use it to alter the appearances of video samples.
>
> **Weakness 5: Weak motivation for TSM.**
>
> Thanks for this question. The reasons for choosing TSM can be summarized into two folds:
>
> (1) The implementation of TSM does not contain other modern data augmentations so it will be easier for us to analyze the efficacy of MCA. In contrast, Uniformer[3] is a competing method that involves multiple strong data augmentations, e.g., Mixup[4]/CutMix[5]/RandAugment[6], and the efficacy of MCA upon these methods will be different from the improvement brought by its own. Nevertheless, we highlight that MCA is compatible with those methods based on our analysis in Fig. 7.
>
> (2) TSM is a 2D network which is based on ResNet50[7] and is efficient in implementation. Considering the resources we have, we choose TSM to save the training time so that we can conduct more experiments to validate the effectiveness and generalization ability of MCA.
>
> To further address the Reviewer's concern, we implement Uniformer, which is a very competitive transformer method, on other datasets for empirical validation:
> | Method  | Mini-Kinetics Acc. | HMDB51 Acc. |  UCF101 Acc. |
> | :-----: | ------------ | :-----: | :-----: |
> | Uniformer | 76.12% | 45.75% | 82.45% |
> | Uniformer+MCA | 77.18% | 50.20% | 84.27% |
>
> It can be noticed that MCA consistently boosts the performance of Uniformer across different datasets which validates its effectiveness and generalization ability.
>
> **Weakness 6: Channel Swap has been defined.**
>
> Thanks for pointing it out. First, we want to clarify that Channel Swap is only part of the design of SwapMix. Further, we emphasize that our motivation is different from theirs as Channel Swap is proposed to simulate the effect of hue variance in video recognition and there is no prior work have done so based on our best knowledge.

---

> ### Author Response · Authors · 2023-11-15
> **Response to Reviewer 7MY2 (part 3)**
>
> **Weakness 7: Definition of Diversity and ECE.**
>
> Thanks for bringing it out and we will add the definitions of Diversity and Expected Calibration Error (ECE) in our final version.
>
> **Weakness 8: Difference between Fig.5 and Tab.5.**
>
> We thank the Reviewer for the careful reading. In Tab. 5, the model is trained on augmented data and evaluated on clean validation data like normal supervised learning. While the results in Fig. 5 are obtained by evaluating the pre-trained model TSM, which is trained on clean data, on different validation sets, where the data is original videos and video samples that are generated by SwapMix and Hue Jittering. This figure corresponds to the definition of Affinity in Sec 3.1 and we want to prove that SwapMix leads to less performance deline compared to Hue Jittering which means it has higher Affinity and suffers less from distribution shift.
>
> ***
>
> We thank the Reviewer again for the suggestions which help us to improve the work. We are actively available until the end of this rebuttal period and let us know if you have any further questions. Looking forward to hearing back from you.
>
>
> [1] Lin J, Gan C, Han S. Tsm: Temporal shift module for efficient video understanding[C]//Proceedings of the IEEE/CVF international conference on computer vision. 2019: 7083-7093.
> [2] Wang J, Gao Y, Li K, et al. Removing the background by adding the background: Towards background robust self-supervised video representation learning[C]//Proceedings of the IEEE/CVF Conference on Computer Vision and Pattern Recognition. 2021: 11804-11813.
> [3] Li K, Wang Y, Gao P, et al. Uniformer: Unified transformer for efficient spatiotemporal representation learning[J]. arXiv preprint arXiv:2201.04676, 2022.
> [4] Zhang H, Cisse M, Dauphin Y N, et al. mixup: Beyond empirical risk minimization[J]. arXiv preprint arXiv:1710.09412, 2017.
> [5] Yun S, Han D, Oh S J, et al. Cutmix: Regularization strategy to train strong classifiers with localizable features[C]//Proceedings of the IEEE/CVF international conference on computer vision. 2019: 6023-6032.
> [6] Cubuk E D, Zoph B, Shlens J, et al. Randaugment: Practical automated data augmentation with a reduced search space[C]//Proceedings of the IEEE/CVF conference on computer vision and pattern recognition workshops. 2020: 702-703.
> [7] He K, Zhang X, Ren S, et al. Deep residual learning for image recognition[C]//Proceedings of the IEEE conference on computer vision and pattern recognition. 2016: 770-778.

---

> > ### Comment · Area_Chair_PV3V · 2023-11-19
> >
> > Dear Reviewer,
> >
> > The author has provided responses to your questions and concerns. Could you please read their responses and ask any follow-up questions, if any?
> >
> > Thank you!

---

> > ### Comment · Reviewer_7MY2 · 2023-11-20
> > **Clarification of "appearance"**
> >
> > Thank for the responce with additional experiments. I have read the feedback, and I am still not convinced what the "static appearance" is that the paper addresses, and how it relates to background bias.
> >
> > In this paper, appearance refers to the color of the pixels in the scene, and swap-mix mixes RGB pixel values with blend. But "appearance" includes more than that, such as texture, shape, and background scene information. It is a strong bias, and the CLIP paper (Radford+, 2021) showed performance of UCF101 and Kinetics400, which is much higher than this paper, with a linear probe using only a single frame (it is static). Therefore, the papers listed in the comment have tried to mitigate the bias of the background, which contains the "appearance" of the scene. The authors repeat that "static appearances are less important in videos" in the text and response, but the bias prevents methods from ignoring static scene/background information, and the proposed swap-mix doesn't seem to solve the problem.

---

> ### Author Response · Authors · 2023-11-20
> **Reply to Reviewer 7MY2**
>
> Dear Reviewer 7MY2,
>
> Thank you for the feedback. It seems that the reviewer has a different understanding of 'static appearance' with us and we make further explanations to clarify this point.
>
> ***
>
> **Definition of 'appearance'.**
>
> 1. Indeed, we focus on hue variance in this paper which is an important color attribute. As the reviewer mentioned, 'appearance' is a general concept that contains many attributes, such as color and background scene information. Thus, **modifying any of these attributes will lead to appearance changes**.
>
> 2. Based on this dependency (hue variance $\Rightarrow$ color variance $\Rightarrow$ appearance variance), it suggests that our method will lead to appearance changes in videos so that **MCA can force the model to learn appearance invariant representations** (a similar explanation is that Random Rotation will help the model become more robust to changes in orientation).
>
> ***
>
> **Claim of 'static appearances are less important in videos'.**
>
> 1. Our logic is that: MCA $\Rightarrow$ appearance variance in video samples $\Rightarrow$ model learn appearance invariant representation $\Rightarrow$ improved performance. Therefore, we think static appearances are less important in videos because learning appearance invariant representations is beneficial for performance.
>
> 2. Although starting from different perspectives (hue variance/background scene variance), the goal of these two research directions is similar: **learn appearance invariant representations (invariant to hue/background scene) and prioritize the motion information**.
>
> ***
>
> **MCA does not address the background bias issue.**
>
> 1. We sincerely agree with the reviewer that background bias is an essential problem in video recognition. However, our work does not focus on the background bias problem (we have not claimed to address this issue in our paper), and **resolving this issue is not the only way to develop data augmentation methods for video recognition**. For instance, previous image-based data augmentation methods (e.g., RandAugment) do not focus on this issue but they are still effective in video recognition. Yet, MCA is different from these image-based methods as hue variance brings negative effects in object recognition, indicating that our method is specially tailored for video recognition.
>
> 2. To some extent, MCA can partially address the background bias issue as it will also cause hue variance in the background area. Furthermore, compared to the prior background bias works, we can help the model to rely less on the foreground bias information as well. As shown in Fig. 1 in our paper, the action 'catch' remains unaffected despite the hands and football turning into green color with hue variance.
>
> 3. Moreover, **these two lines of research (hue variance/background scene variance) can be combined with each other**. As shown in our reply to Weakness 3, our method is compatible with prior work BE that tries to address the background bias issue, and leads to further improvement when combined with it:
>
> | Method | Top-1 Acc. |
> | :-----: | :-----: |
> | TSM | 45.63% |
> | TSM(BE) | 46.45%(+0.82) |
> | TSM(MCA) | 47.57%(+1.94) |
> | TSM(BE+MCA) | 48.05%(+2.42) |
>
> ***
>
> **Stronger performance on UCF101 and Kinetics400 with CLIP.**
>
> 1. For the stronger results of CLIP, we clarify that CLIP is trained on tons of data with a much stronger backbone ViT-G, while our backbone model is ResNet-50 which is pre-trained on ImageNet.
>
> 2. For the strong bias on UCF101 and Kinetics400, we acknowledge its existence and that's why we conduct most of the experiments on Something-Something V1/V2 datasets which contain strong temporal dependency and methods rely on static appearances achieve inferior performance on them. For example, we compare the results of CLIP, DINOv2, and TSM on Something-Something V2 dataset. It is shown that TSM achieves much better performance and our method can further enhance the performance by 1.4%, which suggests that MCA can encourage the model to prioritize motion information and rely less on static appearances.
>
> | Method | Backbone | Top-1 Acc. |
> | :-----: | :-----: | :-----: |
> | CLIP | ViT-G/14 | 35.8% |
> | DINOv2 | ViT-g/14 | 38.3% |
> | TSM | ResNet50 | 59.3% |
> | TSM+MCA | ResNet50 | 60.7% |
>
> ***
>
> We hope these clarifications can help the reviewer to better understand our work and motivation. We will add this discussion in our final version and we appreciate the Reviewer's efforts for us to improve the work.
>
> We are actively available at this stage and please let us know if you have any further questions.

---

> > ### Author Response · Authors · 2023-11-21
> > **Kind Reminder of Deadline**
> >
> > Dear Reviewer 7MY2,
> >
> > Thanks for your efforts so far for our paper review.
> >
> > Currently, most of the concerns from other Reviewers have been well-addressed and we are eager to know whether our further response has resolved your remaining concerns. Concretely, we have further clarified the meaning of 'appearance' and discussed the relation between MCA and methods that target the background bias issue in our newest response. Moreover, we have updated our draft based on your suggestions by adding the discussion, comparison, and definitions of certain metrics.
> >
> > Due to the coming discussion deadline (11/22), we would like to kindly remind Reviewer 7MY2 if our response through the whole rebuttal has addressed any of your concerns and helped the Reviewer to reevaluate this work.
> >
> > We really appreciate it if you could give us any feedback and your opinions are rather important to us. Thank you very much for your time!
> >
> > Sincerely,
> > Authors

---

### Official Review · Reviewer_4nis · 2023-10-31

**Soundness:** 3 good
**Presentation:** 3 good
**Contribution:** 3 good
**Rating:** 8
**Confidence:** 4

**Summary:**

The paper introduces a new augmentation technique for training video recognition networks, referred to as Motion Coherent Augmentation (MCA). The method is a simple but effective modification of existing Hue Jittering method, adapted specifically to video content (e.g. to learn appearance invariant representation, capturing motion patterns). The method is well explained and the experiments confirm the efficacy of the proposed solution.

**Strengths:**

Simple but effective augmentation technique
Can be easily applied to improve any of video recognition methods
Evaluated on multiple datasets

Overall the paper is well written and clearly presents the contributions. The proposed augmentation technique (MCA) consists of two parts: SwapMix ( a method, which permutes RGB channels and computes the linear interpolation between original and the permuted image), and Variation Alignment (a method that forces the network to become invariant to SwapMix, ie. the network is explicitly trained to return the same class distributions for both, original image and SwapMix-ed image. The multiple experiments on multiple video datasets confirm the efficacy of the proposed technique. The ablation analysis of different components of the method has also been presented.

**Weaknesses:**

Slows down the training and requires more GPU resources
The analysis of impact of \lambda_{AV} has not been evaluated.

**Questions:**

Could authors provide the analysis of \lambda_{AV} impact on training performance?

---

> ### Author Response · Authors · 2023-11-15
> **Response to Reviewer 4nis**
>
> We appreciate the Reviewer's approval and valuable comments. We respond to the Reviewer's concerns as below.
>
> **Weakness 1: Training Overhead.**
>
> Thanks for pointing it out. Indeed, MCA will inevitably increase the training time of the baseline method as other data augmentation methods. However, the training efficiency of MCA can be controlled by the hyperparameter $\rho$ and the analysis can be found in Appendix A.6 of our paper. Notably, MCA requires fewer training hours compared to Hue Jittering when $\rho$ is smaller than 0.75 with a clear advantage in accuracy. Moreover, the training time of MCA varies a lot with different choices of $\rho$ but there is only a small change in accuracy, indicating that we could choose a small $\rho$ if we want to achieve a better trade-off between training efficiency and validation accuracy.
>
> **Weakness 2: Analysis of $\lambda_{AV}$.**
>
> We thank the Reviewer for the great suggestion. We further conduct experiments with different $\lambda_{AV}$ over TSM[1] on Something-Something V1 dataset.
> | Method | $\lambda_{AV}$ | Top-1 Acc. |
> | :-----: | :-----: | :-----: |
> | TSM | - | 45.63% |
> | TSM+MCA | 0.25 | 46.79% |
> | TSM+MCA | 0.5 | 47.40% |
> | TSM+MCA | 1 | 47.57% |
> | TSM+MCA | 2 | 47.03% |
> | TSM+MCA | 3 | 46.57% |
>
> It can be observed that MCA outperforms the baseline method with different choices of $\lambda_{AV}$ and MCA is robust to this hyperparameter variation.
> However, if $\lambda_{AV}$ is too large or small, we observe that the performance will start to decrease which meets our expectation as MCA will degenerate to the baseline if $\lambda_{AV}=0$ and $L_{AV}$ will dominate the training if $\lambda_{AV}$ is too large.
> Due to the short time window during the rebuttal, we only tested several choices of $\lambda_{AV}$ and will conduct a more sufficient analysis in our final version.
>
> ***
>
> Again, we thank the reviewer for the support and valuable advice for us to improve the paper. We are actively available until the end of this rebuttal period and let us know if you have any further questions. Looking forward to hearing back from you.
>
>
> [1] Lin J, Gan C, Han S. Tsm: Temporal shift module for efficient video understanding[C]//Proceedings of the IEEE/CVF international conference on computer vision. 2019: 7083-7093.

---

> > ### Comment · Area_Chair_PV3V · 2023-11-19
> >
> > Dear Reviewer,
> >
> > The author has provided responses to your questions and concerns. Could you please read their responses and ask any follow-up questions, if any?
> >
> > Thank you!

---

### Official Review · Reviewer_SoaL · 2023-11-01

**Soundness:** 3 good
**Presentation:** 3 good
**Contribution:** 3 good
**Rating:** 6
**Confidence:** 3

**Summary:**

The current study investigates the effect of hue variance on video recognition and proposes a data augmentation method called Motion Coherent Augmentation (MCA) that introduces appearance variation to encourage the model to prioritize motion patterns over static appearances. This approach is based on the observation that static appearances are less important in videos with motion information. The proposed method uses an operation called SwapMix to modify video samples' appearances efficiently and Variation Alignment (VA) to resolve distribution shifts caused by SwapMix, enforcing the model to learn appearance-invariant representations. Comprehensive experiments on different architectures and datasets demonstrate the effectiveness and generalization ability of MCA, achieving an average performance gain of 1.95% on the Something-Something V1 dataset compared to the competing method Uniformer.

**Strengths:**

The proposed Motion Coherent Augmentation (MCA) method addresses the issue of overfitting in video recognition by introducing an effective data augmentation strategy. The key advantages of MCA are:

- Hue Jittering: MCA leverages Hue Jittering, which is usually overlooked in object recognition, to generate new appearances for videos. This operation helps the model prioritize the motion pattern over static appearance.

- SwapMix: An efficient operation called SwapMix is introduced to modify the appearance of video samples in the RGB space. This operation helps simulate the effect of appearance variation while preserving important color attributes like saturation and lightness.

- Variation Alignment (VA): VA is used to construct training pairs with different appearances and encourage their predictions to align with each other. This helps the model learn appearance-invariant representations.

- Compatibility: MCA can be seamlessly integrated into existing video recognition approaches with minimal code modifications and provides consistent improvement. This indicates that the method is compatible with existing data augmentation techniques.

Experimental validation: Comprehensive experiments on various architectures and benchmarks demonstrate the effectiveness and generalization ability of MCA. The method demonstrates excellent performance and can further enhance the results of competing approaches like Uniformer.

In summary, the Motion Coherent Augmentation method addresses the issue of overfitting in video recognition by providing an effective data augmentation strategy that leverages Hue Jittering, SwapMix, and Variation Alignment. MCA achieves consistent improvement while being compatible with existing techniques, making it a promising approach for video recognition.

**Weaknesses:**

It is a valid point that the paper lacks a discussion on the generalization capabilities of MCA and how it could potentially improve the transferability of models. Introducing such a discussion could indeed help increase the impact of the work. A possible approach could be to analyze the performance of MCA on tasks that are different from the benchmark datasets used during training, and compare it to other data augmentation methods. Additionally, analyzing the learned representations using techniques like clustering or visualization could provide insights into their generalization potential. Overall, a thorough evaluation of the generalization capabilities of MCA could strengthen the paper and make it more impactful.

The relationship between MCA and existing video self-supervised methods is not further explored in the given text. It would be valuable to investigate how MCA can be integrated with other video self-supervised methods and examine the potential synergies between them. Understanding how MCA complements or enhances existing techniques could provide valuable insights into the effectiveness and generalization abilities of the combined approach. Further research is needed to explore the relationship between MCA and video self-supervised methods and determine how they can be effectively integrated to improve video recognition tasks.

**Questions:**

Whether the MCA data augmentation method provides gains in video domain adaptation or video domain generalization settings, where videos from different domains are involved, is not explicitly discussed in the given context. To assess whether MCA is beneficial in such video migration scenarios, an evaluation of its performance in tasks that involve domain adaptation or generalization would be necessary. Comprehensive experiments comparing MCA to other data augmentation techniques and assessing its effect on domain adaptation or generalization metrics would provide insights into its potential usefulness in video migration settings.


The potential of integrating MCA with existing video self-supervised methods to further enhance the performance is not explicitly discussed in the given text. However, it is worth exploring the compatibility of MCA with other video self-supervised methods and investigating whether their combination can lead to improved results. By combining the strengths of MCA in introducing appearance variation and prioritizing motion patterns with other self-supervised techniques, it is possible to enhance the overall effectiveness of video representation learning. Further research and experimentation are needed to explore the potential synergies and benefits of integrating MCA with existing video self-supervised methods.

---

> ### Author Response · Authors · 2023-11-15
> **Response to Reviewer SoaL**
>
> We thank the Reviewer for delivering valuable comments to help us improve our work. We make the response as below.
>
> **Weakness 1: Lack of discussion on generalization and transferability.**
>
> We thank the Reviewer for the valuable suggestion. In order to prove that the representation learned by MCA can be generalized to other domains, we first pre-train TSM[1] on Kinetics400 with different data augmentation methods and then conduct linear probing on the HMDB51 dataset to compare the downstream performance.
> | Method  |  Top-1 Acc. |
> | :-----: | :-----: |
> | TSM |  63.59% |
> | TSM+AugMix |  64.18% |
> | TSM+RandAugment |  65.10% |
> | TSM+MCA |  65.95% |
>
> It can be noticed that MCA results in the best performance compared to other methods, which proves that the representation learned by MCA is general and can be transferred to other domains.
>
> **Weakness 2: Relation with Self-Supervised Learning.**
>
> Thanks for the great suggestion. To demonstrate the effectiveness of MCA in self-supervised learning, we implement MoCo[2] on UCF101 dataset and further combine it with MCA. Specifically, we conduct pre-training and fine-tuning both on the UCF101 dataset following the setting of BE[3] and utilize the 3D network I3D[4] as the base encoder. Given the short time window during rebuttal, we reduce the resolution during pre-training to $112\times112$ and increase the batch size for efficiency.
> | Method | Top-1 Acc. |
> | :-----: | :-----: |
> | MoCo | 63.68% |
> | MoCo+MCA | 67.99% |
>
> One can observe that MCA leads to significant improvement over MoCo on the UCF101 dataset which proves the efficacy and generalization ability of MCA in different video recognition tasks and we will add this part in the final version.
>
> ***
>
> We thank the Reviewer again for the suggestions which help us to improve the work. We are actively available until the end of this rebuttal period and let us know if you have any further questions. Looking forward to hearing back from you.
>
>
> [1] Lin J, Gan C, Han S. Tsm: Temporal shift module for efficient video understanding[C]//Proceedings of the IEEE/CVF international conference on computer vision. 2019: 7083-7093.
> [2] He K, Fan H, Wu Y, et al. Momentum contrast for unsupervised visual representation learning[C]//Proceedings of the IEEE/CVF conference on computer vision and pattern recognition. 2020: 9729-9738.
> [3] Wang J, Gao Y, Li K, et al. Removing the background by adding the background: Towards background robust self-supervised video representation learning[C]//Proceedings of the IEEE/CVF Conference on Computer Vision and Pattern Recognition. 2021: 11804-11813.
> [4] Carreira J, Zisserman A. Quo vadis, action recognition? a new model and the kinetics dataset[C]//proceedings of the IEEE Conference on Computer Vision and Pattern Recognition. 2017: 6299-6308.

---

> > ### Comment · Area_Chair_PV3V · 2023-11-19
> >
> > Dear Reviewer,
> >
> > The author has provided responses to your questions and concerns. Could you please read their responses and ask any follow-up questions, if any?
> >
> > Thank you!

---

### Official Review · Reviewer_sFSx · 2023-11-06

**Soundness:** 3 good
**Presentation:** 3 good
**Contribution:** 2 fair
**Rating:** 6
**Confidence:** 2

**Summary:**

This paper presents an interesting trick to improve action recognition performance in benchmark datasets. Specifically, hue jittering or hue variance has been shown to help improve performance most likely due to adding robustness to the appearance of the content and focusing more on the motion to perform better action recognition. The proposed MCA approach focuses on better data augmentation for video recognition tasks. Performance on multiple benchmarks shows evidence for this idea to be effective.

**Strengths:**

Proposes a simple yet effective trick to improve the quantified results by a small amount on multiple benchmark datasets.

Comparison performed on multiple action recognition benchmark datasets with ablation study as well.

The writing, figures, and explanation are reasonable and effective.

**Weaknesses:**

The novelty of the approach is limited, as-in a new network to extract unique embedding features is not what is being proposed here. This seems more like add-on that could be used with other solutions.

**Questions:**

1. How does this approach compare to the one that uses only optical flow or other motion representation and not appearance?
2. Does the MCA approach work better with other more modern and robust core networks than TCM?

---

> ### Author Response · Authors · 2023-11-15
> **Response to Reviewer sFSx**
>
> We appreciate the Reviewer’s approval and constructive suggestions for us to improve our work. We make the response as below.
>
> **Weakness 1: Limited novelty.**
>
> Thanks for the comment. Indeed, our method is simple yet effective, as our aim is to deliver a method that has strong generalization ability and can be seamlessly integrated into existing approaches to improve their performance (shown in Tab. 1/2, Fig. 6/7). Nevertheless, we want to emphasize our contribution as:
>
> (1) We have strong motivation: Hue Jittering is often neglected in current video recognition methods and we find this behavior is beneficial as static appearances are less important in videos. This justifies why MCA can lead to improvement on different datasets, architectures, and other data augmentation methods, as this problem generally exists in video recognition and is often neglected in current methods.
>
> (2) We propose effective techniques: the current implementation of Hue Jittering is highly time-consuming and its efficacy is limited by the distribution shift problem. To solve these two issues, we propose SwapMix to simulate the effect of hue variance while being significantly faster in practice, and then we present Variation Alignment to resolve the distribution shift and implicitly encourage the model to focus more on the motion information.
>
> (3) We conduct comprehensive validation: extensive experiments validate the effectiveness and generalization ability of MCA over different architectures and benchmarks, and its compatibility with existing data augmentation techniques.
>
>
> **Weakness 2: Comparison with optical flow or motion representation methods.**
>
> Thanks for the good question. Similar to other data augmentation methods, MCA is designed for RGB modality and it is hard to fairly compare the results from different modalities as they exhibit very different performance. To validate the efficacy of MCA, we revise the self-supervised learning method BE[1], which is robust to background bias and prioritizes motion information, to data augmentation operation and adapt it in supervised learning where we train it on Something-Something V1 dataset over TSM[2] for fair comparisons:
> | Method | Top-1 Acc. |
> | :-----: | :-----: |
> | TSM | 45.63% |
> | TSM(BE) | 46.45%(+0.82) |
> | TSM(MCA) | 47.57%(+1.94) |
> | TSM(BE+MCA) | 48.05%(+2.42) |
>
> As shown in this table, MCA clearly outperforms BE and is compatible with BE which leads to significant improvement of baseline method. This further demonstrates the effectiveness and generalization capability of MCA and we will add the results in the final version.
>
>
> **Weakness 3: Effectiveness on other modern networks.**
> Thanks for asking this question. We validate MCA over different architectures in Tab. 2 where it is implemented on the popular 3D network: SlowFast[3] and very competing Transformer network: Uniformer[4] and MCA consistently improves the accuracy of these methods. Further, we stress that Uniformer is a competitive modern network that is composed of multiple data augmentations and exhibits very strong performance. From our analysis in Fig. 6/7, MCA is compatible with those augmentation operations and can further boost its performance across different frame numbers.
> | Method | Frame | Top-1 Acc. | GFLOPs |
> | :-----: | :-----: | :-----: | :-----: |
> | Uniformer-S | 32 | 58.8% | 329 |
> | Uniformer-S+MCA | 20 | 59.3%(+0.5) | 169 |
>
> Notably, Uniformer-S (20 Frame) with MCA obtains the accuracy of 59.3% on Something-Something V1 dataset, outperforming the official results Uniformer-S (32 Frame) with higher accuracy and markedly less computational costs.
>
> ***
>
> Again, we thank the Reviewer for the constructive suggestions which help us to improve the work. We are actively available until the end of this rebuttal period and let us know if you have any further questions. Looking forward to hearing back from you.
>
>
> [1] Wang J, Gao Y, Li K, et al. Removing the background by adding the background: Towards background robust self-supervised video representation learning[C]//Proceedings of the IEEE/CVF Conference on Computer Vision and Pattern Recognition. 2021: 11804-11813.
> [2] Lin J, Gan C, Han S. Tsm: Temporal shift module for efficient video understanding[C]//Proceedings of the IEEE/CVF international conference on computer vision. 2019: 7083-7093.
> [3] Feichtenhofer C, Fan H, Malik J, et al. Slowfast networks for video recognition[C]//Proceedings of the IEEE/CVF international conference on computer vision. 2019: 6202-6211.
> [4] Li K, Wang Y, Gao P, et al. Uniformer: Unified transformer for efficient spatiotemporal representation learning[J]. arXiv preprint arXiv:2201.04676, 2022.

---

> > ### Comment · Area_Chair_PV3V · 2023-11-19
> >
> > Dear Reviewer,
> >
> > The author has provided responses to your questions and concerns. Could you please read their responses and ask any follow-up questions, if any?
> >
> > Thank you!

---

> > ### Comment · Reviewer_sFSx · 2023-11-23
> > **Response reviewed**
> >
> > Thank you for providing additional details and clarifications. In light of the interesting insights shared by all reviews and follow-on discussions, I think I would like to maintain my original rating of the paper.

---

### Author Response · Authors · 2023-11-21
**Updated Draft**

Dear Reviewers:

Thanks for your valuable comments made in the review process. We have revised the draft based on your suggestions and the revised area is marked in yellow color. Specifically, we have:
***
* added the self-supervised learning results in Appendix A.2,
* validated the transferability of MCA in Appendix A.3,
* studied the hyperparameter effect in Appendix A.4,
* added the comparisons with VideoMix and BE in Tab.3,
* discussed with background scene variation methods in Sec.2 and A.5,
* added the definition of Diversity and Expected Calibration Error (ECE).
***

Please let us know if you have any remaining concerns and thank you so much for being with us so far.

Sincerely,
Authors

---

### Meta-Review · Area_Chair_PV3V · 2023-12-06

**Metareview:**

The paper investigates the effect of hue variance in video action recognition and finds it effective for learning motion from videos. Subsequently, the paper proposes the Motion Coherent Augmentation (MCA) method based on this observation.

After taking into account the reviews, the responses from the authors, and reading the paper myself, AC recommends acceptance with a poster. AC strongly suggests the authors incorporate all the feedback and additional experiments required from the reviewers in the final version.

Strengths and weaknesses are summarized below:

Strengths:
1. the method is simple and yet very effective
2. rigorous experiments have verified the effectiveness of the approach (especially after the rebuttal)
3. the paper is well-written and easy to follow

Weaknesses:
1. as pointed out by reviewer 7MY2, though the paper does not directly address the problem of background biases, it would be interesting to discuss these similar augmentation approaches in the related works, highlight the differences, and discuss the potentials of the proposed method in removing such biases in the future work section.
2. as pointed out by reviewer SoaL, it is also important to discuss the generalization/transfer ability of the model trained with the proposed augmentation method in future works.

**Justification For Why Not Higher Score:**

The paper does not make significant contributions to qualify for a spotlight or oral, as the applicability of the method is rather narrow. It is unknown whether it is not generalizable to other tasks, such as image recognition, or eliminates other biases in video action recognition, such as background biases.

**Justification For Why Not Lower Score:**

The paper proposes a very simple yet elegant method to improve motion perception in video action recognition. This augmentation method has been less studied in the past in video action recognition. This simple and effective method can be very useful for real-world practices.

---

### Decision · Program_Chairs · 2024-01-16

Accept (poster)